# Halogen Bonding and CO-Ligand Blue-Shift in Hybrid Organic—Organometallic Cocrystals [CpFe(CO)$_2$X] (C$_2$I$_4$) (X = Cl, Br)

**Yury Torubaev** [1,*] **, Ivan Skabitskiy** [1]**, Sergey Shapovalov** [1]**, Olga Tikhonova** [1] **and Anna Popova** [2]

[1] Kurnakov Institute of General and Inorganic Chemistry, Russian Academy of Sciences, 119991 Moscow, Russia; skabitskiy@gmail.com (I.S.); schss@yandex.ru (S.S.); olga_tikhonova12@mail.ru (O.T.)

[2] The Faculty of Physics, Mathematics and Natural Sciences, Peoples' Friendship University of Russia (RUDN), 117198 Moscow, Russia; a.popova1701@gmail.com

\* Correspondence: torubaev@igic.ras.ru

**Abstract:** This work is focused on the complex interplay of geometry of I⋯X halogen bonds (HaB) and intermolecular interaction energy in two isomorphic cocrystals [CpFe(CO)$_2$X] (C$_2$I$_4$) (X = Cl (**1**), Br (**2**)). Their IR-spectroscopic measurements in solid state and solution demonstrate the blue-shift of CO vibration bands, resulting from I⋯X HaB. The reluctance of their iodide congener [CpFe(CO)$_2$I] to form the expected cocrystal [CpFe(CO)$_2$I] (C$_2$I$_4$) is discussed in terms of different molecular electrostatic potential (MEP) of the surface of iodide ligands, as compared with chloride and bromide, which dictate a different angular geometry of HaB around the metal-I and metal-Br/Cl HaB acceptors. This study also suggests C$_2$I$_4$ as a reliable HaB donor coformer for metal-halide HaB acceptors in the crystal engineering of hybrid metal–organic systems.

**Keywords:** halogen bonding; σ-hole; cocrystal; blue-shift



## 1. Introduction

Two functional groups in tetrahaloethenes, C$_2$X$_4$ (X = F, Cl, Br, I), namely double bonds and halogen, provide at least two possible types of their coordination in transition metal complexes: the π-coordination of C=C double bond and donor–acceptor interaction between the halogen atom and metal centre. Both these types are exemplified by a sufficient number of crystal structures deposited in the Cambridge Structural Database (CSD). Each category has its certain specificity: C=C double bond coordination is known only for rather compact tetrafluoroethene (C$_2$F$_4$) with platinum group metals (Pt, Pd, Rh), while I → M donation was found only in tetraiodide C$_2$I$_4$ complexes with Ag and Au [1,2]. Recent advances in the investigation of non-covalent interactions and their use in supramolecular design revealed two more types of intermolecular interactions between C$_2$I$_4$ and metal complexes: "regular" halogen bonding (HaB) between iodine functions of C$_2$I$_4$ and HaB-accepting halide ligands in metal complexes, investigated by the same group [Pt$_2$I$_6$] [3], Ni bis(2-oxido-*N,N*-dimethyldiazene-1-carboximidamide) [4] and direct metal-to-iodine donation, or I⋯M halogen bonding (HaB), developed by the group of V. Kukushkin [5]. The former type of "regular" HaB-assisted hybrid metal–organic cocrystals can be considered as a part of a big family of HaB cocrystals between C$_2$I$_4$ and halide salt HaB acceptors such as [R$_4$N]$^+$X$^−$, developed mostly by the efforts of the group of W. Pennington [6]. Unlike the metal-halide supramolecular architectures, such [R$_4$N]$^+$X$^−$ C$_2$I$_4$ salt cocrystals feature more than two (typically four, rarely six) molecules of C$_2$I$_4$ in the "coordination sphere" of the halide anion. We were interested in investigating whether such big "coordination numbers" can be achieved in metal-halide cocrystals of C$_2$I$_4$. On the other hand, in a continuation of our research of the HaB effect on the electronic structure of transition metal complexes [7], we were interested in the investigation of the crystal and electronic structure of the hybrid

metal–organic cocrystal of transition metal carbonyls and $C_2I_4$. Therefore, we cocrystallized $CpFe(CO)_2X$ (X = Cl, Br, I) with $C_2I_4$ and studied the crystal structure, geometry, energy of intermolecular interactions, and IR spectra of the resulting cocrystals.

## 2. Materials and Methods

### 2.1. Materials

Solvents were purified, dried, and distilled under an argon atmosphere before use. Commercial $C_2I_4$ was used without additional purification. The $[CpFe(CO)_2X]$ (X = Cl, Br, I) cocrystals were prepared using reported procedures [Brauer].

### 2.2. General Co-Crystallization Procedure

Cocrystals **1–2** were prepared by the slow evaporation of $CH_2Cl_2$ solution of coformers.

In a typical experiment, samples of cocrystal **1** suitable for single-crystal XRD investigation were grown in 1.5 mL plastic vial (Eppendorf) containing a solution of $[CpFe(CO)_2X]$ (0.1 mmol) and of $C_2I_4$ (0.2 mmol) in 0.25 mL of $CH_2Cl_2$, which was closed and left at room temperature for 48 h. Red crystals of **1** along with some colorless prisms of $C_2I_4$ formed. They were washed with 1 mL of hexane, then dried in a stream of argon.

### 2.3. X-ray Structure Determinations

A Bruker D8 Venture diffractometer (Bruker, Billerica, MA, USA) equipped with graphite-monochromated Mo K$\alpha$ radiation (0.71070 Å) was used for the cell determination and intensity data collection for cocrystals **1** and **2** and $CpFe(CO)_2Br$. The data were collected by the standard 'phi-omega scan techniques and were reduced using SAINT v8.37 A (Bruker: Madison, WI, USA). The SADABS (version 2016/2; Bruker, 2016) software was used for scaling and absorption correction. The structures were solved by direct methods and refined by full-matrix least-squares against $F^2$ using Olex2 and SHELXTL software [8,9]. Non-hydrogen atoms were refined with anisotropic thermal parameters. All hydrogen atoms were geometrically fixed and refined using a riding model.

### 2.4. Computations

Theoretical calculations were carried out with the ORCA 5.02 program package [10]. A non-hybrid PBE functional [11] dispersion correction with Becke–Johnson damping (D3BJ) [12,13] and a def2-TZVP basis set [14] with small-core pseudopotential for I atoms [15] were used for geometry optimization. Def2/J auxiliary basis [16] was used for Coulomb fitting. Electron density calculations of the resulting geometries were performed using ZORA approximation for scalar relativistic effects [17,18], a hybrid functional PBE0 [19] and an all-electron def2-TZVP basis set reconstructed for ZORA. RIJCOSX approximation [20] in combination with a SARC/J auxiliary basis set [21] was used to improve computational speed. MEP values on $0.002\ e\text{Å}^{-3}$ isosurfaces were evaluated by the Multiwfn program [22]. MEP figures were prepared using VisMap software [23].

Intermolecular interaction energy calculation and subsequent energy framework generation were performed in Crystal Explorer 17.5 (TONTO, B3LYP-DGDZVP) for all unique molecular pairs in a molecule's first coordination sphere (4 Å) using experimental crystal geometries.

## 3. Results

### 3.1. Preparation and Crystal Structure of [CpFe(CO)$_2$X]·(C$_2$I$_4$) (X = Cl, Br)

Cocrystallization of 2:1 $C_2I_4$ and $CpFe(CO)_2X$ solution in $CH_2Cl_2$ in the conditions of slow evaporation at room temperature afforded isomorphic 1:1 cocrystals of $[CpFe(CO)_2Cl]·(C_2I_4)$ (**1**) and $[CpFe(CO)_2Br]·(C_2I_4)$ (**2**), respectively. A first glance at their crystal structure reveals the $CpFe(CO)_2X$ molecule connected with four $C_2I_4$ molecules by $I\cdots X$ HaBs (Figure 1). In turn, each iodine in $C_2I_4$ molecule forms $I\cdots X$ HaB, so that $C_2I_4$ molecule is surrounded by four $CpFe(CO)_2X$ molecules (see Figure 2).

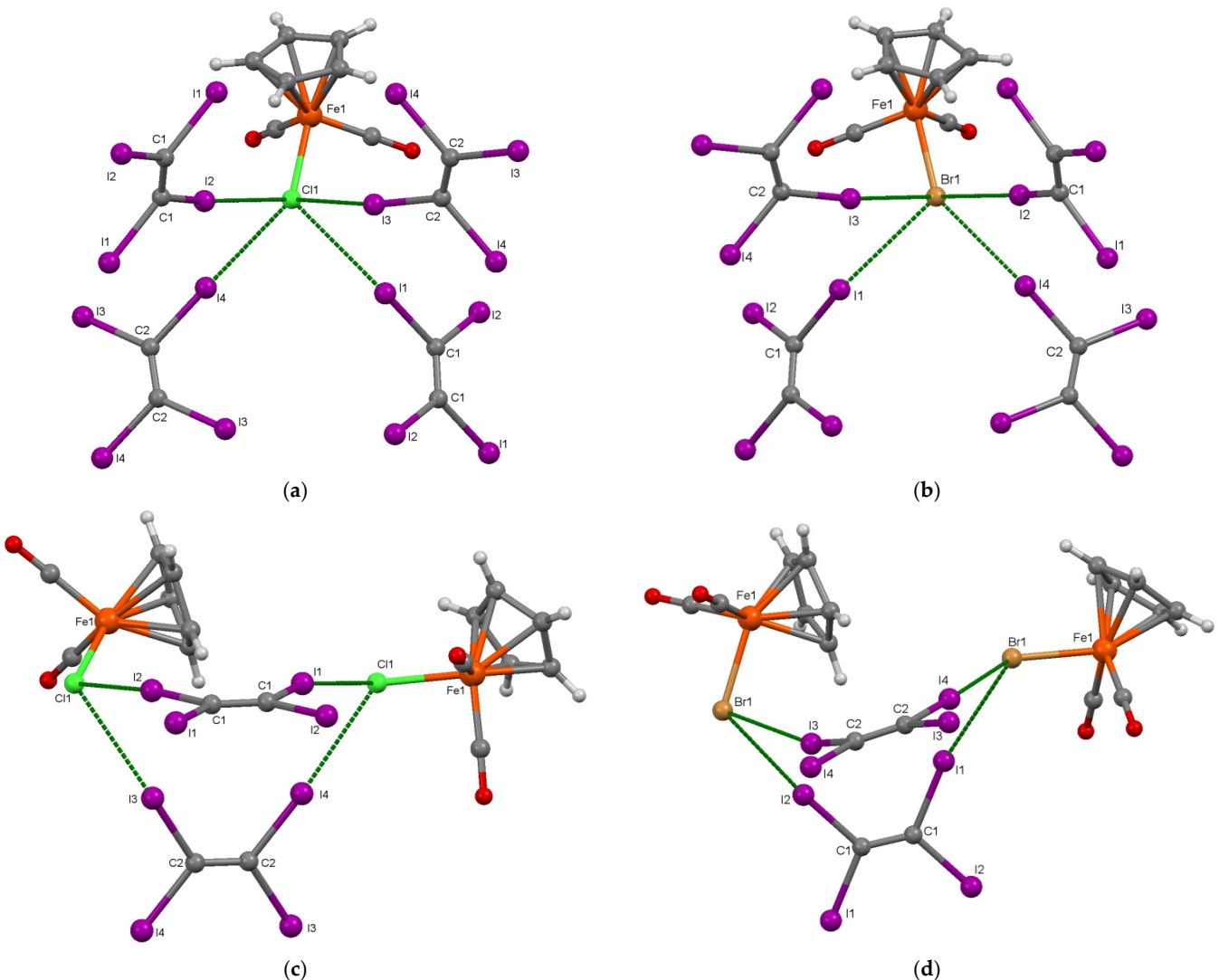

**Figure 1.** Fragments of the crystal structure of **1** (**a,c**) and **2** (**b,d**) showing the HaB-assisted supramolecular arrangement around [CpFe(CO)$_2$X] molecules (**a,b**) and two independent C$_2$I$_4$ molecules, bridging two [CpFe(CO)$_2$X] (**c,d**). Selected interatomic distances in **1** (Å): Fe1-Cl1 2.295(2), Cl1···I1 3.443(2), I3···Cl1 3.382(2), Cl1···I2 3.259(2), Cl1···I4 3.261(2); Selected angles in **1** (°): Fe1-Cl1···I1 114.06(8), Fe1-Cl1···I2 113.40(8), Fe1-Cl1···I3 100.64(8), Fe1-Cl1···I4 130.31(9), C1-I2···Cl1 168.7(3), C1-I1···Cl1 166.4(3), C2-I3···Cl1 162.8(3), C2-I4···Cl1 175.7(3). Selected interatomic distances in **2** (Å): Fe1-Br1 2.430(1), Br1···I1 3.5598(7), Br1···I2 3.3717(7), Br1···I3 3.4517(8), Br1···I4 3.3398(7). Selected angles in **2** (°): Fe1-Br1···I1 112.56(3), Fe1-Br1···I2 112.13(3), Fe1-Br1···I3 97.03(3), Fe1-Br1···I4 134.93(3), C1-I1···Br1 164.3(2), C1-I2···Br1 170.2(2), C2-I3···Br1 161.3(2), C2-I4···Br1 176.0(2).

We can note a slight elongation of the Fe-Br bond in **2** (2.430(1) Å) as compared with 2.4221(4) Å in the native [CpFe(CO)$_2$Br], and Fe-Cl bond (2.295(2) Å) in **1,** compared with 2.251(7) Å in [CpFe(CO)$_2$Cl], trapped in a cyclodextrine cage [24].

Tetrahaloethenes C$_2$I$_4$ and C$_2$Br$_4$ tend to reproduce the layered fragments of their native packing patterns (i.e., in their parent crystal) in their cocrystals such that two of three lattice dimensions of the native C$_2$X$_4$ (X = Br, I) and their pyrazine adducts are similar [25–27]. However, close examination of crystal structures **1** and **2** did not reveal any parallels with the crystal structure of the native C$_2$I$_4$. The σ-holes at each of four iodine atoms of C$_2$I$_4$ in **1** and **2** are directed towards the halide ligands X of CpFe(CO)$_2$X, forming I···X HaBs (Figure 2); therefore, short I···I interactions (HaBs) between C$_2$I$_4$ molecules are absent in these cocrystals. Each halide ligand of CpFe(CO)$_2$X in turn forms four I···X

HaBs (Figure 1). The lengths of these I···X HaBs vary in the range of 3.259(2) Å–3.443(2) Å in **1** to 3.3398(7) Å–3.5598(7) Å in **2**, but in general, they are shorter than the respective sums of vdW radii, and their Fe-X···I angles suggest that all of these HaBs are genuine (type-II [28]) HaBs.

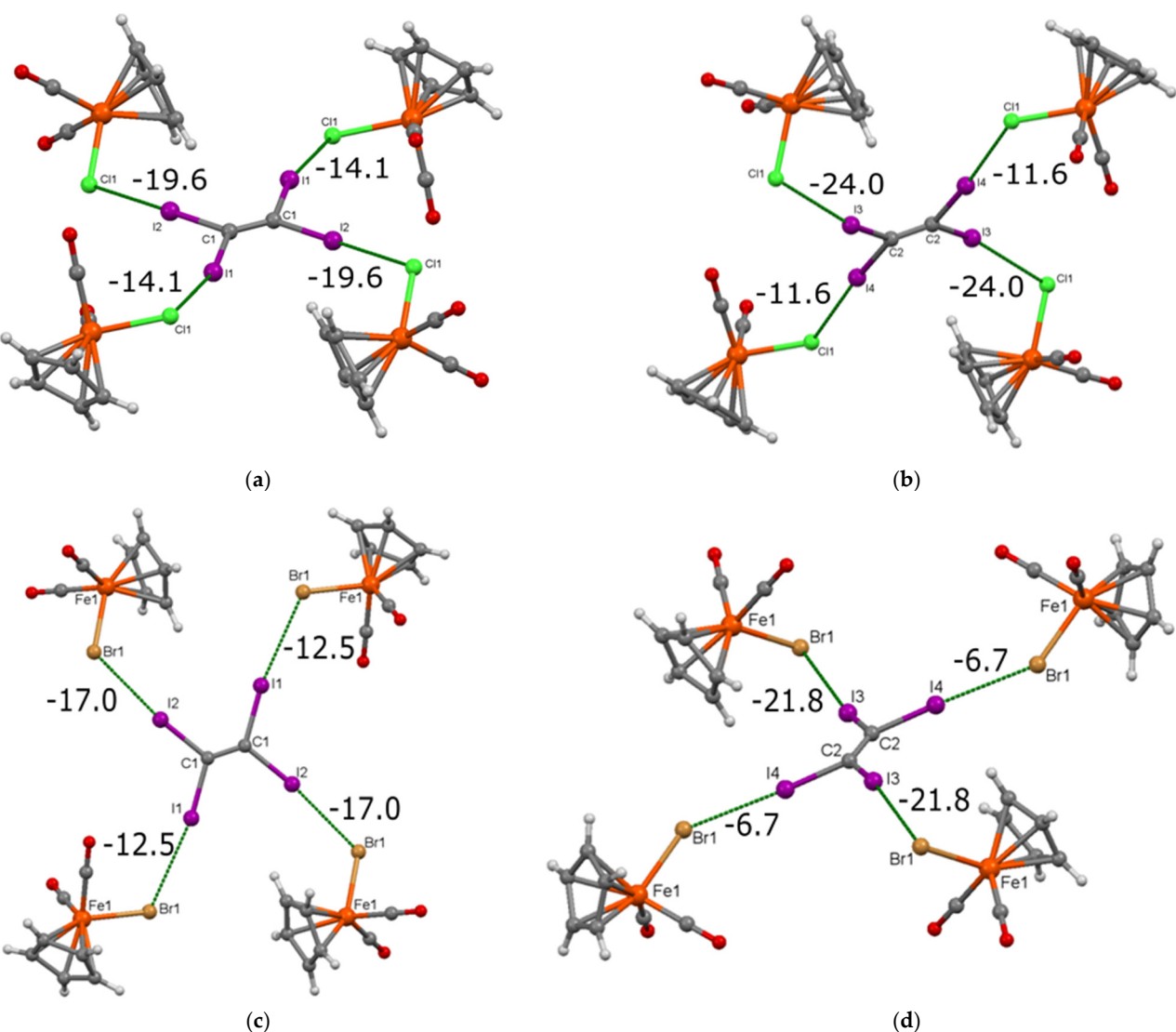

(a)  (b)

(c)  (d)

**Figure 2.** Fragments of the crystal structure of **1**, showing the supramolecular arrangement and intermolecular interaction energies around two independent $C_2I_4$ molecules in **1** (**a**,**b**) and in **2** (**c**,**d**). Energy values are indicated in kJ/mol. Note that these are the total intermolecular interaction energies of the given molecular pairs, but not specific I···X HaB energies.

As a final structural remark, we can mention that in the same conditions as for $CpFe(CO)_2X$ (X = Cl, Br), the $CpFe(CO)_2I$ did not afford any cocrystalline product with $C_2I_4$ and only starting materials were recovered from the reaction mixture. The reluctance of $CpFe(CO)_2I$ in terms of the supramolecular interaction with $C_2I_4$ is most likely due not to the weak properties of the iodide ligand as an acceptor compared with chloride and bromide, but to its increased demands on the angles at which stable I···X-Fe HaB can form. As we demonstrated earlier [7], in some cases (against the background of a fairly stable and flexible I···Cl-M and I···Br-M HaBs), the formation of I···I-M HaB was not observed precisely because of steric factors—the impossibility of placing a HaB donor in the equatorial plane (of the nucleophilic p-belt) of the iodine ligand due to the sterically crowded environment around it. For the same reason, the HaB-assisted cocrystals

of metal-chloride and metal-bromide HaB donors are usually isomorphic, while metal-I cocrystals with the same HaB donors stay apart [29]. We can assume that since the Fe1-X1···I4 angles in **1** and **2** (130° and 135°, respectively, Figure 1) are clearly above the right angle and 106°–113° observed before in [Bu$_4$P][Pt$_2$I$_6$] C$_2$I$_4$ cocrystal [3], the same arrangement for CpFe(CO)$_2$I and C$_2$I$_4$ is not stable. Furthermore, we can speculate that other arrangements for CpFe(CO)$_2$I and C$_2$I$_4$ may not provide sufficient profit in energy and packing as compared with the starting materials. Apart from the higher rigidity of the iodide-centered HaBs and consequent steric problems of achieving the optimal angles, another contribution to the failure of CpFe(CO)$_2$I to form cocrystals with C$_2$I$_4$ may come from the lower HaB acceptor properties of the iodide as compared with bromide and chloride. In particular, the series of studies on halopiridinium salts with halide anions as HaB acceptors demonstrated that although the nature of the halide anion provides a secondary effect on the HaB strength, the HaBs with iodide are generally weaker than those with bromide and chloride as HaB acceptors [30–35].

### 3.2. Electronic Structure and IR Spectroscopy

Molecular electrostatic potential (MEP) mapping of C$_2$I$_4$ shows a rather pronounced and equivalent σ-hole on each iodine atom ($V_{max}$ = −36.5 kcal/mol, Figure 3).

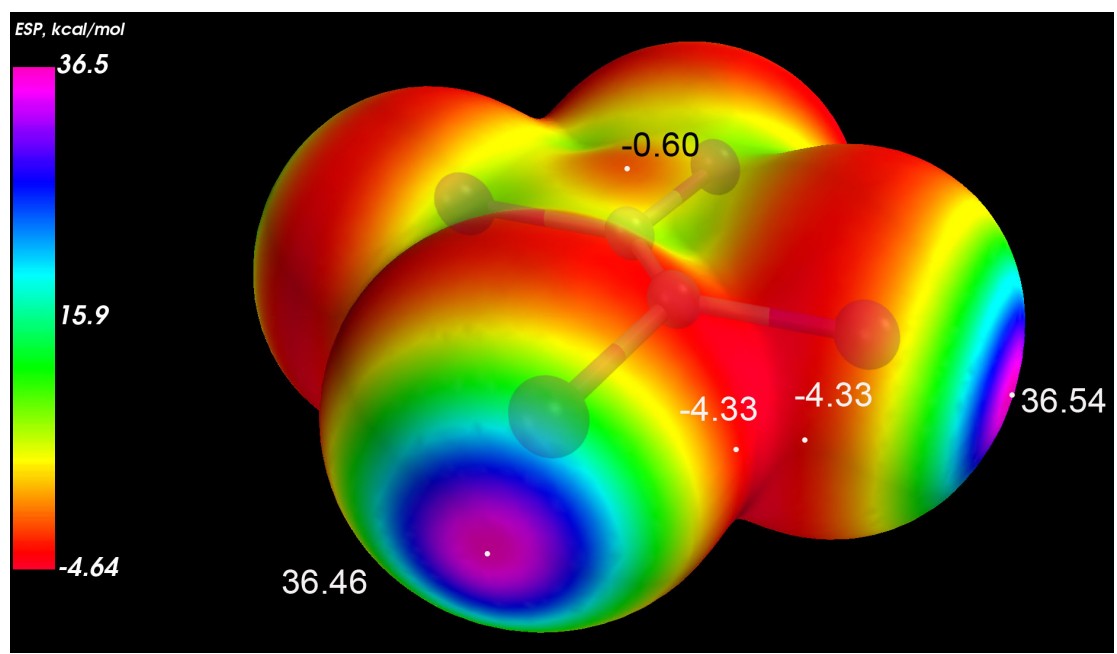

**Figure 3.** Molecular electrostatic potential (MEP) maps for an isolated molecule of C$_2$I$_4$ (0.002 e/A$^3$ electron density isosurface).

The σ-holes in C$_2$I$_4$ are only 5 kcal less "deep" than in 1,4-diiodotetrafluorobenzene (1,4-DITFB, $V_{max}$ = −40.4 kcal/mol [36]), the classic HaB donor, which was successfully used as a coformer for CpFe(CO)$_2$X (X = Cl, Br, I, TePh) in our earlier study [7].

HaB between the positive area of iodine function of C$_2$I$_4$ and the nucleophilic area of ligand X in CpFe(CO)$_2$X is sufficient to shift the electron density from the M-X bond and therefore to reduce M→CO back-donation [37]. The resulting depopulation of the π*C-O LUMO orbital slightly strengthens the CO bond and consequently shifts its vibrational band to the higher frequency area. This is also in good agreement with the slight elongation of Fe-X bonds in **1** and **2** compared with the free molecules of [CpFe(CO)$_2$X] mentioned in Section 3.1.

The series of [CpFe(CO)$_2$X] (1,4-DITFB) (X = Cl, Br, I, TePh) cocrystals demonstrated the 5–10 cm$^{-1}$ blue shift of CO vibrational bands as compared with the respective parent

CpFe(CO)$_2$X. Therefore, the 5–7 cm$^{-1}$ blue shifts of CO vibration bands in the solid-state ATR IR spectra of **1** and **2** (as compared with the parent CpFe(CO)$_2$Cl and CpFe(CO)$_2$Br) were quite expected (Figure 4).

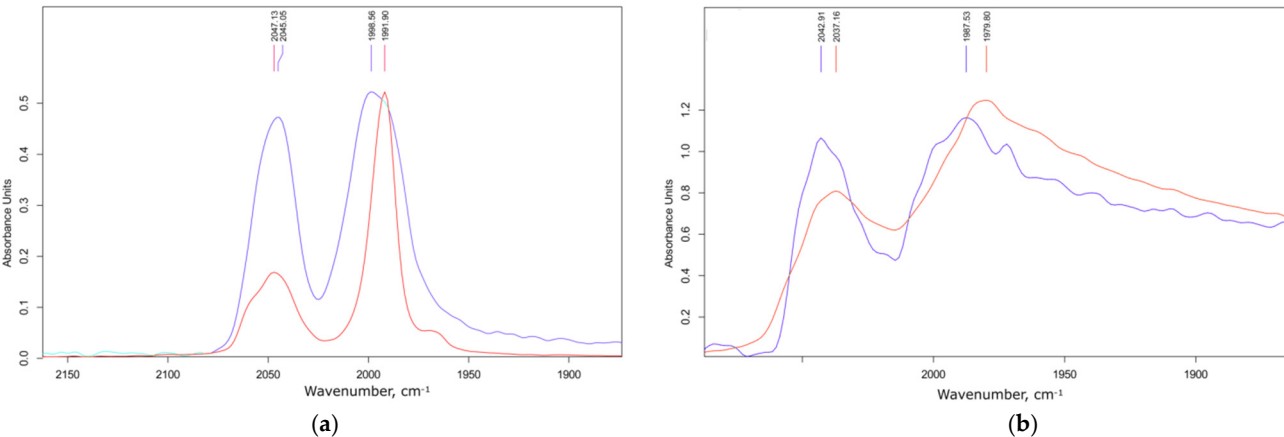

**Figure 4.** Overlay of the fragments of IR (ATR) spectra of pure CpFe(CO)$_2$Cl (red) and cocrystal **1** (blue, **a**); overlay of CpFe(CO)$_2$Br (red) and **2** (blue, **b**).

Such less pronounced blue shifts of CO groups in **1** and **2** are in good agreement with the longer I···Br distances in **2** (3.3398(7)–3.5598(7)Å (Figure 1) as compared with 3.294–3.311 Å in [CpFe(CO)$_2$Br] (1,4-DITFB) [7].

We evaluated the energy of intermolecular interactions between CpFe(CO)$_2$Br and C$_2$I$_4$ in **2** (−12.5−−17.0 kJ/mol) and [CpFe(CO)$_2$Br]–1,4-DITFB (−14.6−−15.0 kJ/mol) in the respective cocrystal in Crystal Explorer/TONTO–DGDZVP [38]. These values reflect the total intermolecular interaction energy, but not specific (e.g., HaB) interactions. Intermolecular interactions between C$_2$I$_4$ and CpFe(CO)$_2$Br molecules in cocrystal **2** are not limited to I···Br HaBs and also include C-H···I contacts. The latter may provide a noticeable contribution to the intermolecular adhesion, and therefore the shorter I···Br distances (say, I2···Br1 3.3717(7) Å) in combination with the H8···I2 contacts (3.1730) correspond to stronger intermolecular bonding (−17 kJ/mol), while the I···Br HaB match the longest I···Br separation (I1···Br1 3.5598(7) Å) and weaker intermolecular interaction (−12.5 kJ/mol). It should be noted that I···Br distances in **2** are the result of a complex interplay of intermolecular interactions and cannot be directly compared with those in [CpFe(CO)$_2$Br] (1,4-DITFB), so we can only generally conclude that the HaB interaction of the bromide ligand in [CpFe(CO)$_2$Br] with four C$_2$I$_4$ is comparable to that with two 1,4-DITFBs.

## 4. Conclusions

C$_2$I$_4$ is a reliable HaB donor coformer for the organometallic halides, particularly for CpFe(CO)$_2$X (X = Cl, Br). The strong demand of the iodide ligand to accept the HaB strictly in the equatorial plane results in the reluctance of CpFe(CO)$_2$I to cocrystallize with C$_2$I$_4$. The formation of four I···X HaBs around each X ligand results in 5–7 cm$^{-1}$ blue shifts of CO vibration bands in the solid-state ATR IR spectra of [CpFe(CO)$_2$X] C$_2$I$_4$ (X = Cl, Br) cocrystals.

**Author Contributions:** Conceptualization, methodology and manuscript preparation, Y.T. and I.S.; XRD crystallography and QC computations, synthesis and spectroscopic measurements, O.T., A.P. and S.S. All authors have read and agreed to the published version of the manuscript.

**Funding:** This research was funded by the Ministry of Science and Higher Education of the Russian Federation, grant number 075-15-2020-779.

**Data Availability Statement:** Atomic coordinates and other structural parameters of **1-x** have been deposited with the Cambridge Crystallographic Data Centre: CCDC 2154590 (**1**), CCDC 2154589 (**2**), and CCDC 2154591 [CpFe(CO)$_2$Br].

**Acknowledgments:** We thank the reviewers for their valuable comments.

**Conflicts of Interest:** The authors declare no conflict of interest. The funders had no role in the design of the study; in the collection, analyses, or interpretation of data; in the writing of the manuscript, or in the decision to publish the results.

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
