# Peer review of "Halogen Bonding and CO-Ligand Blue-Shift in Hybrid Organic—Organometallic Cocrystals [CpFe(CO)2X] (C2I4) (X = Cl, Br)"

_crystals, doi:10.3390/cryst12030412_

Round 1
Reviewer 1 Report
The manuscript 'Halogen Bonding and CO-ligand blue-shift in Hybrid Organic - 2 Organometallic Cocrystals [CpFe(CO)2X] (C2I4) (X=Cl, Br)' by Torubaev et al. is an interesting and clearly written study of structural and spectroscopic effects of halogen bonding in cocrystals of organometalic halogen bond acceptors with tetraiodoethene as halogen bond donor. I would definitely recommend its publication, after some very minor revisions.
- I feel that in their discussion of reasons of failure of formation of the cocrystal with the iodide derivative (lines 130-150), the authors are too quick to dismiss the lower HaB acceptor properties of the iodide as a contributing factor. While their arguments for the higher rigidity of the HaB-s with iodide ligand as an acceptor and steric impossibility of achieving the optimal angles are convincing, there have been numerous studies on 'free' halides as halogen bond acceptors which demonstrate that halogen bonds with iodide are considerably weaker than those with bromide and chloride as acceptors (see e.g. New J. Chem. 1999, 23 (12), 1137–1139; Zeitschrift fur Naturforsch. - Sect. B J. Chem. Sci. 2001, 56 (9), 889–896; J. Phys. Chem. A 2007, 111 (12), 2319–2328; CrystEngComm 2020, 22 (23), 4039–4046; Crystal Growth & Design 2021, 21, 11, 6044-6050; Crystal Growth & Design2021, 21, 12, 6889-6901; Crystal Growth & Design 2022, 22, 2, 1333-1344...) With this in mind, I would suspect that both the higher angular rigidity and overall lower C-I...I-M HaB energy contributed to the failure of cocrystallisation. I would therefore suggest the authors to somewhat expand this part of the discussion.
- The halogen bonding contact is sometimes in the text denoted by three central dots between the donor and the acceptor, and sometimes by three hyphens – it would be preferable if dots were always used.
- There are several typos in the text which would require correcting (line 33: ' Ni_bis(2-oxido-N,N-dimethyldiazene-1-carboximidamide)' – the underline is superfluous and 'N'-s should be italic; line 92 ' each iodine in C2I4 molecule form I---X HaB' – '...forms...'; line 125 ' Each halide ligands...' – '...ligand...'; line 126 ' The length of these' – '...lengths...'; line 152 ' Vmax -36.5 kcal/mol' – 'V' should be italic and '=' is missing (same in 158); line 179 ' Crystal Explore' – 'Explorer'
- There are some errors in the references, in particularly among the paper titles (probably due to copying and pasting). E.g. ref 3 ' Hexaiododiplatinate(ii)' (‘ii’ in lowercase); ref 25 ' CpFe(CO)2Cl.' ('2' not in subscript); ref 29 ' halogen.cntdot..cntdot..cntdot.halogen', ref 30 ' Cp2MX2' (both '2' not in subscript)... Also, ref 24 ends with '(accessed on' (the date is missing)
Author Response
Dear Reviewer,
Thank you very much for your careful examination of our manuscript and helpful comments. We have acknowledged all your suggestions and the detailed responses are listed below. Corrections and changes are highlighted in yellow in the revised manuscript. These corrections, changes and additions have really improved the manuscript and I hope it is now suitable for publication.
Sincerely,
Yury Torubaev
Comment: I feel that in their discussion of reasons of failure of formation of the cocrystal with the iodide derivative (lines 130-150), the authors are too quick to dismiss the lower HaB acceptor properties of the iodide as a contributing factor. While their arguments for the higher rigidity of the HaB-s with iodide ligand as an acceptor and steric impossibility of achieving the optimal angles are convincing, there have been numerous studies on 'free' halides as halogen bond acceptors which demonstrate that halogen bonds with iodide are considerably weaker than those with bromide and chloride as acceptors (see e.g. New J. Chem. 1999, 23 (12), 1137–1139; Zeitschrift fur Naturforsch. - Sect. B J. Chem. Sci. 2001, 56 (9), 889–896; J. Phys. Chem. A 2007, 111 (12), 2319–2328; CrystEngComm 2020, 22 (23), 4039–4046; Crystal Growth & Design 2021, 21, 11, 6044-6050; Crystal Growth & Design2021, 21, 12, 6889-6901; Crystal Growth & Design 2022, 22, 2, 1333-1344...) With this in mind, I would suspect that both the higher angular rigidity and overall lower C-I...I-M HaB energy contributed to the failure of cocrystallisation. I would therefore suggest the authors to somewhat expand this part of the discussion.
Response: Thank you for this valuable comment. Yes, we agree that not only steric, but electronic effects of halide HaB acceptors should be discussed, and so have added suggested references with the respective comment on the lower HaB acceptor properties of the iodide as another contributing factor.
Comment: The halogen bonding contact is sometimes in the text denoted by three central dots between the donor and the acceptor, and sometimes by three hyphens – it would be preferable if dots were always used.
Response: Corrected.
Comment: There are several typos in the text which would require correcting (line 33: ' Ni_bis(2-oxido-N,N-dimethyldiazene-1-carboximidamide)' – the underline is superfluous and 'N'-s should be italic; line 92 ' each iodine in C2I4 molecule form I---X HaB' – '...forms...'; line 125 ' Each halide ligands...' – '...ligand...'; line 126 ' The length of these' – '...lengths...'; line 152 ' Vmax -36.5 kcal/mol' – 'V' should be italic and '=' is missing (same in 158); line 179 ' Crystal Explore' – 'Explorer'
Response: Corrected
Comment: There are some errors in the references, in particularly among the paper titles (probably due to copying and pasting). E.g. ref 3 ' Hexaiododiplatinate(ii)' (‘ii’ in lowercase); ref 25 ' CpFe(CO)2Cl.' ('2' not in subscript); ref 29 ' halogen.cntdot..cntdot..cntdot.halogen', ref 30 ' Cp2MX2' (both '2' not in subscript)... Also, ref 24 ends with '(accessed on' (the date is missing)
Response: Corrected.
Reviewer 2 Report
Comments
The article describe fir the first time the Br type CpFe(CO)2 structure as well as two structures with tetra iodate ethylene adducts. The structures are relatively well solved in spite of problems with crystals. The structures are discussed in terms of intermolecular interactions caused by electrostatic potential on iodine in terms of sigma-holes. Paper is rather routine, synthesis are simple, but paper is of general interest and is well prepared. The figures are of good quality, paper lengths is appropriate and literature is correct. Some corrections are necessary, as indicated below. After this rather cosmetic changes paper can be accepted.
Corrections
- Please add the solvent name and not only DCM as this is not familiar shortening for all.
- 4. and text above it. Authors wrote CpFe(CO)2X. Shouldn’t it be CpFe(CO)2Br? Do spectra of X = I are identical with that of Br analogue? If not, Authors should directly indicate which complex was measured. I think that spectrum 4(a) is for X = Cl and 4(b) for X = Br but there is no any information?
- The spectrum at Fig. 4 (a). In figure caption, the spectrum of CpFe(CO)2X should be in blue and is red-violet and not blue as in (b). Please correct. In spectrum (b) the shift is blue (to higher energy), but in (a) it seems to be red (if spectrum of 2 is really red).
- The structures indicate that both in 1 and 2 there are no any H-bonds. But in text (page 6) Authors include such bonds. The Cp H atoms are not enough close to suggest such an interactions, moreover the angles are < 150o, thus are very very weak.

Author Response
Dear Reviewers,
Thank you very much for your careful examination of our manuscript and helpful comments. We have acknowledged all your suggestions and the detailed responses are listed below. Corrections and changes are highlighted in yellow in the revised manuscript. These corrections, changes and additions have really improved the manuscript and I hope it is now suitable for publication.
Sincerely,
Yury Torubaev
Comment: Please add the solvent name and not only DCM as this is not familiar shortening for all.
Response: Corrected
Comment: 4. and text above it. Authors wrote CpFe(CO)2X. Shouldn’t it be CpFe(CO)2Br? Do spectra of X = I are identical with that of Br analogue? If not, Authors should directly indicate which complex was measured. I think that spectrum 4(a) is for X = Cl and 4(b) for X = Br but there is no any information?
Response: Corrected.
Comment: The spectrum at Fig. 4 (a). In figure caption, the spectrum of CpFe(CO)2X should be in blue and is red-violet and not blue as in (b). Please correct. In spectrum (b) the shift is blue (to higher energy), but in (a) it seems to be red (if spectrum of 2 is really red).
Response: Corrected. The parent CpFe(CO)2X are now red and their cocrystals 1 nad 2 are blue.
Comment: The structures indicate that both in 1 and 2 there are no any H-bonds. But in text (page 6) Authors include such bonds. The Cp H atoms are not enough close to suggest such an interactions, moreover the angles are < 150o, thus are very very weak.
Response: Yes, these H-bonds are indeed very weak and perhaps they should not be called H bonds, but H---I contacts. This is corrected. We didn’t calculate the energies of particular interatomic interactions in this work, and discuss only total intermolecular interaction energy, but from our previous detailed studies, we can suggest that even such weak interactions may contribute to the intermol interaction energy and should be taken into account. Without exact values this sounds speculative, I agree, so we are ready to remove this fragment of discussion. However, its revised form seems more reasonable now.